

# NEAL: an open-source tool for audio annotation

Anthony Gibbons[1], Ian Donohue[2], Courtney Gorman[2], Emma King[2] and Andrew Parnell[1]

[1] Hamilton Institute, Department of Mathematics and Statistics, Maynooth University, Kildare, Ireland
[2] Zoology, School of Natural Sciences, Trinity College Dublin, Dublin, Ireland

## ABSTRACT

Passive acoustic monitoring is used widely in ecology, biodiversity, and conservation studies. Data sets collected via acoustic monitoring are often extremely large and built to be processed automatically using artificial intelligence and machine learning models, which aim to replicate the work of domain experts. These models, being supervised learning algorithms, need to be trained on high quality annotations produced by experts. Since the experts are often resource-limited, a cost-effective process for annotating audio is needed to get maximal use out of the data. We present an open-source interactive audio data annotation tool, NEAL (Nature+Energy Audio Labeller). Built using R and the associated Shiny framework, the tool provides a reactive environment where users can quickly annotate audio files and adjust settings that automatically change the corresponding elements of the user interface. The app has been designed with the goal of having both expert birders and citizen scientists contribute to acoustic annotation projects. The popularity and flexibility of R programming in bioacoustics means that the Shiny app can be modified for other bird labelling data sets, or even to generic audio labelling tasks. We demonstrate the app by labelling data collected from wind farm sites across Ireland.

# INTRODUCTION

Passive acoustic recording is now a staple of ecological monitoring (*Ross & Allen, 2014*; *Hagens, Rendall & Whisson, 2018*; *Sugai et al., 2018*; *Rogers et al., 2013*). Remote sensors can collect vast quantities of high quality audio data at a cost affordable to both academic researchers and citizen scientists. However, the volume of data collected can quickly surpass feasible manual labelling ability, and automatic methods are actively being developed (*Morgan & Braasch, 2021*; *Brunoldi et al., 2016*; *Baumgartner et al., 2019*).

Application of deep learning, in particular convolutional neural networks (CNNs) (*Lecun et al., 1998*), to image processing for audio classification is growing in both popularity and effectiveness (*Hershey et al., 2017*). High performing models have been produced for bat classification (*Mac Aodha et al., 2018*), insects (*Yin et al., 2021*), aquatic mammals (*Thomas et al., 2019*); and bird calls, ranging from relatively few (*Stowell et al., 2019*) to tens (*Salamon et al., 2017*) to even thousands (*Kahl et al., 2021*) of classes. While custom CNNs can be

Corresponding author
Anthony Gibbons,
anthony.gibbons.2022@mumail.ie

trained from scratch, a lack of large *labelled* bioacoustic datasets (*Baker & Vincent, 2019*) can impede model performance.

In recent years, audio classification models have benefited from Transfer Learning (*Ntalampiras, 2018*; *Zhong et al., 2020*). This is where a new model is created with the assistance of a neural network pre-trained using a relatively large dataset for a similar task (such as another audio classification *Hershey et al., 2017* or even image classification *Simonyan & Zisserman, 2014*). The new model adjusts the predictions of the larger model by performing further training with the small amount of labelled data available for the task of interest. In both the custom CNN and transfer learning settings, some training data are still required to tune to the specific application area, and routine test data should be annotated to monitor performance over time. For novel tasks such as medical imaging classification, manufacturing defect detection, agricultural yield prediction and marine image classification, domain specialists are often needed to label the initial training data and evaluate the model output over time (*Sarma et al., 2021*; *Ng, 2021*; *Srivastava et al., 2022*; *Langenkämper et al., 2019*).

Here, we focus on bird species detection, which requires experts to manually label files so that they can be input into a supervised learning algorithm. We present an audio annotation tool that aims to reduce the bottlenecks associated with audio annotation, improving the efficiency of the expert's time, which is often at a premium, and providing finer granularity of labelled data (time-frequency limits, species, call type, additional notes) so multiple classification tasks can be carried out on the same data simultaneously.

In much of the existing audio labelling software, supplemental information important to decision making, such as the exact time of the recording, general location, geographic coordinates and shortest distance to the coastline are often not readily available to the user, reducing the effectiveness of the application as a decision tool. Giving annotators this temporal and spatial information can help contextualise hard-to-classify sound clips. In relatively complicated soundscapes, such as wind farms or urban environments, users often lack the flexibility to temporarily filter out noise and focus on the sound of interest, further increasing labelling time. We include the ability to display various metadata and two methods of filtering audio in our app.

We present NEAL (Nature+Energy Audio Labeller), an interactive Shiny app designed for audio data annotation. It allows users to visually and audibly interpret audio files and label any sounds observed and offers time and frequency granularity for precise labeling. The app incorporates metadata to inform labellers, as well as labeller confidence for each annotation to provide quality annotations.

Some of the key strengths of NEAL are that:

- it is primarily intended to be used locally but can also be deployed to a server;
- it has automatic frequency filtering of selected areas of the spectrogram to remove unwanted sounds during analysis;
- it displays clear metadata for each audio file, giving context on the geography, habitat and time of year recordings were taken;
- labellers can specify label confidence behind each annotation, as opposed to each individual classification (*e.g.*, species of a bird vocalisation) being assumed to have 100% confidence associated with it;
- users can search through existing labels and navigate to those of interest;
- user annotations can be downloaded in bulk.

A comparison between NEAL and popular existing tools for bioacoustic annotation projects (*Audacity Team, Copyright 1999-2021*; *Marsland et al., 2019*; *Fukuzawa et al., 2020*; *Cannam, Landone & Sandler, 2010*) is shown in Table 1. While the Shiny app was built to have a user-friendly layout for manual annotation of bird vocalisations with the option of local or server-side use, it does not yet have vast functionality in terms of bulk classification or training custom species classification models in-app.

The Shiny app NEAL (Fig. 1) is presented together with open-source code and sample data to allow for further modification or deployment. Whilst our focus during development was on bird call detection, the app can be easily changed to enable labelling of other audio tasks. This facilitates the adoption of the Shiny app in projects where complicated soundscapes and data quality may differ greatly among sites and equipment.

We provide the overall layout of a labelling project on the Shiny application, as well as a brief overview of the procedures involved and some of the computational workarounds to avoid computation waiting time. We then demonstrate a workflow of classifying bird species on wind farm sites across Ireland, with step-by-step directions of how the app is utilized. We expand on the input and output formats of the data to allow custom projects to be established easily. Source code for the NEAL App, as well as a link to a working demo on an RStudio server, is available at https://github.com/gibbona1/neal. Portions of this text were previously published as part of a preprint (https://arxiv.org/abs/2212.01457).

## METHODS

### User Interface (UI)

The app was built in R (*R Core Team, 2022*) using the Shiny (*Chang et al., 2021*) framework. Shiny is itself a package in R. No knowledge of HTML, CSS, or JavaScript is necessary to build a simple application in Shiny, but small amounts were used here to enhance certain features. One of the many benefits of the Shiny framework is that its end-users do not need any knowledge of R programming to interact with the data and provide annotations. Shiny has already been used in developing ecology-related apps and decision-support tools, such as in species-habitat modeling (*Wszola et al., 2017*), conservation management (*Pascal et al., 2020*) and forest structure assessment (*Silva et al., 2022*).

NEAL makes use of several open-source R packages (*Attali, 2021*; *Bailey, 2022*; *Pedersen et al., 2021*; *Chang, 2021*; *Chang & Borges Ribeiro, 2021*; *Granjon, 2021*; *Perrier, Meyer & Granjon, 2022*; *Aden-Buie, 2022*; *Littlefield & Fay, 2021*; *Silva, 2021*; *Ligges et al., 2018*; *Sueur, Aubin & Simonis, 2008*; *Wickham, 2016*; *Garnier et al., 2021*; *Chang, Luraschi & Mastny, 2020*; *Schloerke, 2020*; *Wickham et al., 2021*; *Wickham, 2019*; *Firke, 2021*; *Xie, Cheng & Tan, 2021*). The most notable are:

**Table 1 Comparison of NEAL with other free labelling software.**

| Feature<br>Date released | NEAL<br>2022 | Audacity<br>2000 | AviaNZ<br>2019 | Koe<br>2019 | SonicVisualiser<br>2005 |
|---|---|---|---|---|---|
| Platform(s) compatible | ⊞ ⌘ ◊ | ⊞ ⌘ ◊ | ⊞ ⌘ ◊ | ⊞ ⌘ ◊ | ⊞ ⌘ ◊ |
| Built with | R, JS | C, C++, Python | Python, C | Python, JS | C++, SML |
| Label format (time segments, bounding boxes) | boxes | boxes[1] | boxes | segments | boxes |
| Label confidence slider | ✓ | × | ×[2] | × | × |
| Band-pass filter for selected spectrogram area | ✓ | ×[3] | ✓ | × | × |
| Changeable class list | ✓ | × | ✓[4] | ✓ | × |
| Dynamic class +/- | ✓ | × | ✓ | × | × |
| Site metadata display | ✓ | ✓ | ✓ | ✓ | × |
| In-app label editing | ✓ | ✓ | ✓ | ✓ | ✓ |
| Operates locally | ✓ | ✓ | ✓ | ✓ | ✓ |
| Deployment to server | ✓ | × | × | ✓[5] | × |
| Changeable spectrogram colour palettes | ✓ | ✓ | ✓ | ✓ | ✓ |
| Bulk export annotations | ✓ | ×[6] | ×[7] | ✓ | ×[6] |
| Filter labels by multiple fields | ✓ | × | × | ✓ | × |
| Multiple concurrent (collaborating) users | ×[8] | × | × | ✓ | × |
| Visualise multiple spectrograms (comparison) | × | ✓ | ✓[9] | ✓ | ✓ |
| Bulk classification | × | ×[10] | ✓ | ✓ | ×[10] |
| Analysing sequence structure | × | × | × | ✓ | × |
| Train a species recogniser in-app | × | × | ✓ | × | × |

**Notes.**

[1] Not by default.
[2] Colour codes.
[3] Not by default but there may be plugins available.
[4] New species added will appear in the list.
[5] Also operates on its own server.
[6] Individual *label tracks* for each file can be downloaded but must be named appropriately.
[7] Per-species annotations can be exported from batch processed files.
[8] Multiple users can work on a single server but this can be slow and updates are not immediate.
[9] Quick review mode after batch processing.
[10] Can view multiple files at once but these must be manually labelled.

- *shinyjs*, which allows for custom JavaScript (JS) plugins, giving extended functionality using only a small amount of JS code. This includes toggling pause/play of the embedded audio element using the spacebar, disabling UI elements and reset buttons.
- *shinyBS* contains extra user-interface (UI) objects such as collapsible panels, giving the app a more compact layout. In particular, the metadata panel, label edit and label summary tables are contained within these collapsible panels and can be opened as needed.
- *shinyFiles* for file and directory navigation. This is especially useful for deployments to server where the folder structure may not be as familiar to users. The folder selected using `shinyDirButton` and `shinyDirChoose` is then searched for audio files.
- *shinydashboard* and *shinydashboardPlus* for the dashboard layout, sidebar and header tabs. Moving the less-used settings and widgets to the sidebar and grouping them into collapsible menus reduces clutter in the Shiny app's main body.
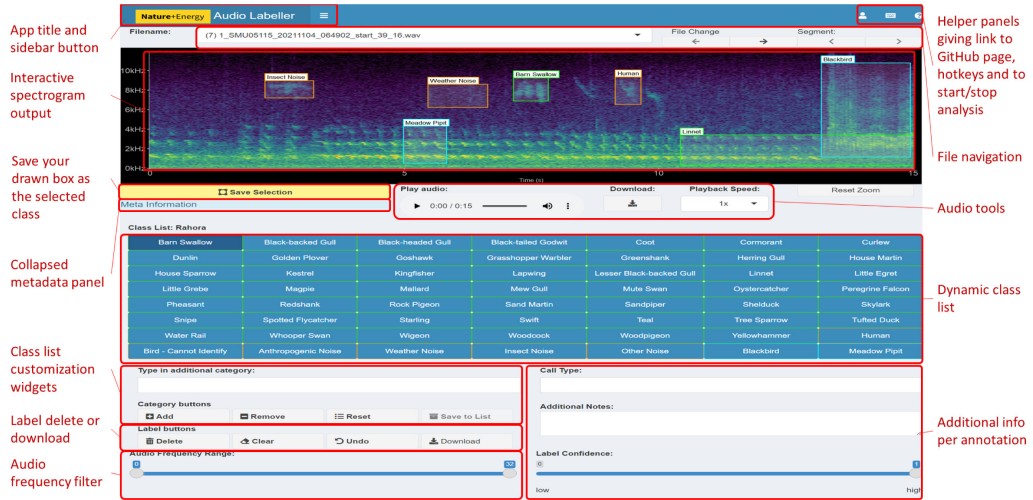

**Figure 1   Main components of the App User Interface.** The main interactive element of the app is the spectrogram, providing a visual representation of the sound. This plot can be drawn on with boxes to filter audio or make annotations. The button on the left underneath the spectrogram is for saving the current selection on the plot, which will be labelled with the class from the grid of categories below. Various audio playback widgets are grouped together to the right of the save button. File navigation allows the user to proceed to the next file in the workflow folder, or select from a drop-down menu of available files, the second of which displays the number of annotations present in each file. The collapsed side panel on the left hand side has extra configuration settings expected to be changed at most once or twice per session. The class list is dynamic: the core (green) categories are selected from a drop-down list in the Configuration tab, while the miscellaneous (orange) categories are static in the current implementation. Custom categories can be added or removed using the category widgets below. Users can provide more information than just the primary category (*e.g.*, bird species) of sound identified; these can include call type, free text additional notes and label confidence.

- *tuneR* for reading and writing audio files. It handles the audio files as `Wave` objects during preprocessing—including dB gain, segmentation and normalization—before passing them to seewave functions.
- *seewave* provides audio waveform manipulation functions, such as spectrogram computation. A noise-reduced or frequency-filtered spectrogram can be reconstructed as a `Wave` object using the `istft` function.
- Graphics are implemented using *ggplot2*. The spectrogram is rendered using `geom_raster` and bounding box annotations are drawn with `geom_rect` and `geom_label`. Additional colour palettes are provided *via viridis*.

A full list of packages employed is available in the source code.

### Display

Upon opening the app, users are presented with the audio data from the chosen file in the form of a spectrogram, which highlights the sound intensity of many frequency levels over time (Fig. 1). Spectrograms are one of the most visually perceptible forms for audio data (*Lin et al., 2012*). The audio file can be played with an embedded audio player underneath the visual.

## General labelling workflow

The standard use of the app is as follows:

① Once the user has opened the app and clicked **start labelling**, the first audio file in their workflow is loaded and the corresponding spectrogram is generated.

② The **spectrogram parameters** such as colour palette and contrast may need to be adjusted until the user is comfortable with the visual distinction of the sounds present.

③ The user plays the audio and, when they come across a sound of interest, they **draw a bounding box** around the vocalisation. This updates the audio player with a **temporary filtered audio file**, keeping only the times and frequencies within the box drawn.

④ Once they have identified the class of sound from the class list, they draw a tight box around the vocalisation and click **save selection**. This will draw a permanent (unless deleted) bounding box labelled with the given class. If they are unhappy with any of the bounding boxes, they can be deleted using the **delete** button in the **label buttons** section.

⑤ If a new class is being added or deleted, or the base species list is changed, this will affect the **class list**—the grid of classes to choose from. This is only a minor computation and the user should not see any delay unless there are several bounding boxes present in the plot.

⑥ The user **continues annotating** the file by repeating steps ② to ⑤ until no more unlabelled sounds of interest remain.

⑦ The user clicks on the arrow to **proceed to the next file** and returns to step ②.

The Shiny framework uses reactive programming, meaning that inputs (in the user interface) changed by the user automatically affect those outputs to which they are connected. This gives a smooth experience for users, where the app does not have to be refreshed manually whenever settings are adjusted. In the case of the app, rendering the spectrogram is the most computationally intensive process. Avoiding the plot refreshing every time an unrelated input is changed is key to a seamless user experience. The back-end of the app includes some modular code to split up dependencies (user inputs), which affect outputs of the user interface. We elaborate on how this code works below.

Figure 2 shows the full layout of the workflow including some of the back-end components. These components are invisible to the user but are required to reduce redundancy in computation. ① to ⑤ correspond to those points in the workflow above.

## Back-end workflow computation
### *Spectrogram plot rendering*

While the computation of spectrograms is relatively fast, primarily due to efficiencies gained by implementing the Fast Fourier Transform (FFT) algorithm (*Brigham & Morrow, 1967*), rendering the result in R using ggplot is slow and creates a noticeable bottleneck for the annotation procedure. As an example, a 15 s audio clip with a sample rate of 24 kHz will produce a spectrogram with dimensions of $128 \times 5{,}622$ when applying the `seewave::spectro` function with an FFT window of 256 points and a window overlap of 75% between two contiguous windows. The `geom_raster` function would then have to plot $\approx 720{,}000$ equally-sized tiles, which is four times that with the linear interpolation
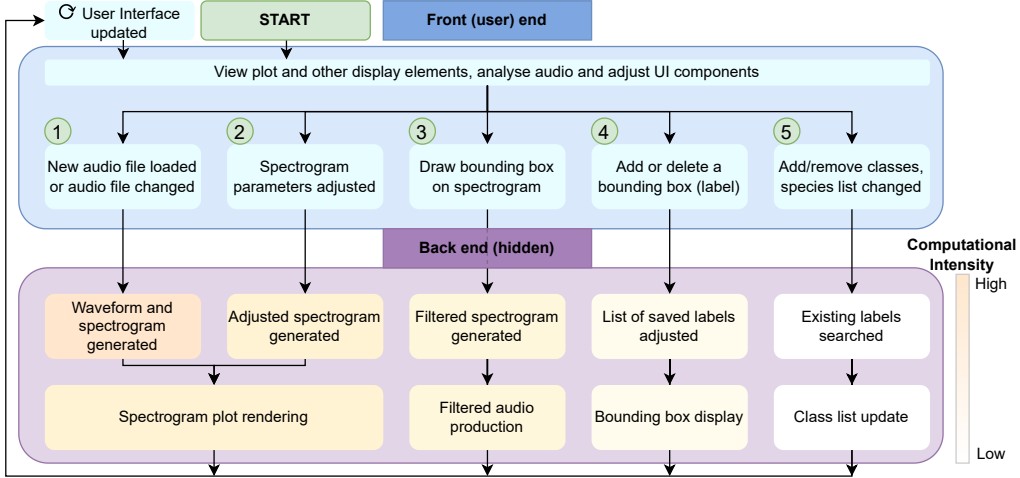

**Figure 2 Workflow diagram.** Demonstration of the front and back-end workflow of the NEAL app. Any analyses or adjustments carried out by the user in the user interface affect the relevant processes in the back end. This keeps the user focused only on the elements being updated while reducing downtime compared to the event that the entire page is refreshed every time. Each of the steps ① to ⑤ in the front end—the actions and changes performed by the user via the UI—have corresponding processes in the background which are in decreasing order of computational demand. The advantage is that these back-end processes have little to no overlap, meaning they can be run separately when the front-end changes are unrelated. If the change is minor (far right, in white) we do not want this to result in an avoidable refresh involving large computations (shown to the left of the figure in orange).

needed for adequate resolution, and then colour by the amplitude (in dB) and apply the chosen colour scale.

This computation must be done at least once for each audio file opened, and may be re-run several times when adjusting FFT parameters, colour palette or zooming in on selected areas of the plot. The common ggplot procedure of adding plot components (such as bounding boxes or frequency guides) to the one plot is not ideal here due to the hindrance caused by frequent interactions with, and thus changes to, the spectrogram plot. If even a small parameter was changed (such as adding a class to the class list UI), the entire plot would have to be re-rendered. Avoiding these barriers to labeller efficiency is paramount to the viability of the Shiny app as an audio annotation tool.

To reduce much of this unnecessary computation, we split the computation work of the spectrogram generation into overlapping panels, shown in Fig. 3. These are often independently generated or refreshed by different widgets in the UI. Figures 3C and 3D have a transparent background and identical axes and padding to the main spectrogram (Fig. 3B). This alignment ensures that all plot layers match exactly.

If the user wants to get a closer look at a particular sound in the spectrogram, a section can be zoomed in on by selecting the area of interest and double clicking on the selected area. The plot should then re-render to show only the selected ranges in time and frequency. The parameters in FFT settings can be tuned further to investigate more complex sounds at this level of magnification. In particular, increasing the FFT window size can put more

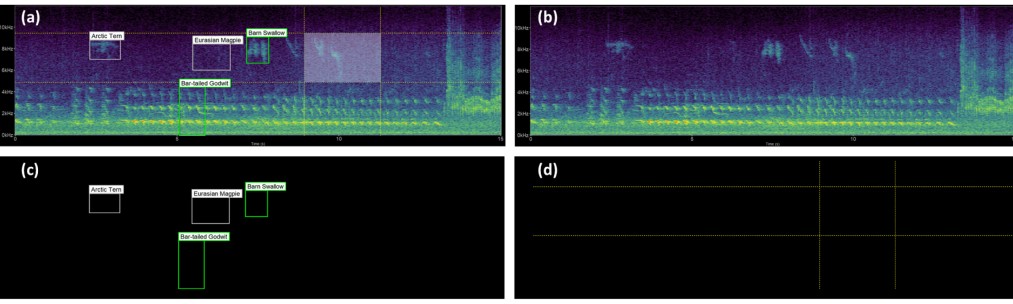

**Figure 3 Spectrogram plot.** The full spectrogram plot (A) is a combination of overlapping panels. From back to front: (B) highlights the main spectrogram, which is the slowest to render; (C) shows the labels panel displaying bounding box annotations; and (D) emphasizes the time/frequency guides which gauge the time and frequency ranges the sound clip occupies, as well as tracing the shape of the bounding box should a label be saved.

emphasis on frequency resolution and less on time resolution. Decreasing this parameter has the opposite effect. To zoom out, the user double clicks any point on the plot, or clicks the Reset Zoom button.

### Filtered audio production

To display the audio to end-users, it is converted to a spectrogram, a visual representation of sound. This is achieved using the Short-Time Fourier Transform (STFT) algorithm (*Allen, 1977*), which applies the FFT algorithm on successive windows of the audio waveform. The output of this transform is a 2D array of complex numbers, before the modulus is taken and the results (scaled to dB) are printed to the screen.

When selecting the audio area of interest, it is often helpful to remove or significantly reduce obvious noise which can hinder clear identification. By default, the app keeps the selected area of the (complex) spectrogram the same, sets all values outside this area to zero (or near-zero) and reconstructs a filtered audio clip using the Inverse Short-time Fourier Transform (ISTFT). To do this, the complex values of the spectrogram are kept in the back-end, rather than just the magnitudes used for visualisation. These complex values are passed into the `seewave::istft` function, based on a MATLAB implementation (*Zhivomirov, 2019*). When the user selects some time/frequency range, only those time/frequency segments will be played. Alternatively, if the audio frequency range slider is adjusted, the entire duration of the audio file is included, but only those frequencies within the filtered range. A similar reconstruction is performed through the spectrogram noise reduction techniques carried out by noise reduction of the complex spectrogram data. Figure 4 illustrates a comparison of the spectrogram and audio player appearances.

The refined audio can often sound unnatural or distorted when the magnitude of the spectrogram values outside the selection are collapsed to zero, so the app has an alternative option to reduce the magnitude of the audio outside the selected region by a factor of 100. The user deselects *Zero Audio outside Selected* in the **Spectrogram Settings**, and when the plot is clicked outside the selected box, the audio is reset.

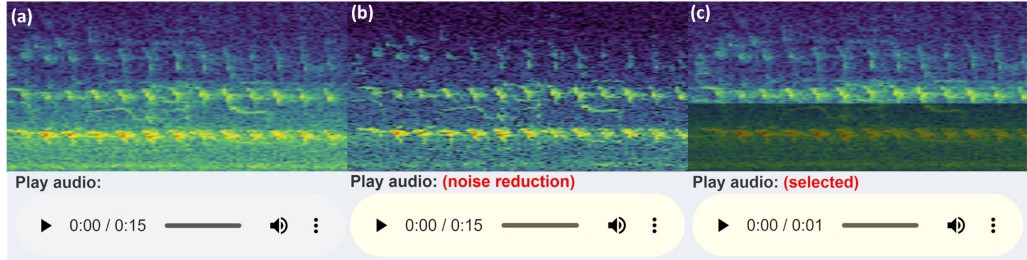

**Figure 4** **Audio Player and spectrogram.** The three versions of the audio UI element the user will see (in Chrome), with an example corresponding spectrogram. The spectrogram has been zoomed in to 0–5 kHz (*y*-axis) and 3–6 s (*x*-axis) to more clearly illustrate the filtering features: (A) shows the default audio player and spectrogram shown to user; (B) displays the audio player and spectrogram with some frequency filtering applied, *i.e.*, the audio within the greyed out box is reduced or removed and the remaining audio is reconstructed; (C) conveys the player and spectrogram when row-wise noise reduction is applied. (A) shows the raw audio file from the workflow folder, while (B) and (C) show the temporary filtered audio files.

### Bounding box display

The annotations created by the NEAL app are bounding boxes: rectangular regions defined using *time* and *frequency* coordinates by the lower left and upper right corners of the object. The purpose of providing a bounding box as the main label is to maximize the detail captured in a sound event annotation project. Figure 5 describes the varying definitions of sound classification projects, based on the diagram in *Stowell (2022)*.

Classifying the sound clip as a whole (Fig. 5A) gives no indication of the duration or regularity (how often it occurs) of a sound event, or whether multiple species are present in a single audio file. Annotation setups where labels transcribe the time dimension with either no frequency resolution (Fig. 5B) or with a small number of set frequency bins is called sound event detection. This does not have the flexibility to account for overlapping vocalisations in the frequency dimension, or a large number of classes where the frequency bins are difficult to distinguish.

Time- and frequency-specific bounding boxes (Fig. 5C) therefore give millisecond level timestamps (both start and end) for sound events, as well as maximum, minimum and median frequencies of the labelled sound clip. If not used directly for object detection (*Zsebők et al., 2019*), frequency metrics can serve as auxiliary input to train machine learning models for species detection (*Lostanlen et al., 2019*).

### Class list update

The class list is a group of buttons representing the available classes to label the identified sounds in the audio files. A predefined list of classes is important for consistency within and among labellers, reducing the time taken to repetitively type class names, as well as minimizing the need to correct typos during post hoc analysis.

The list is displayed in CSS `grid` (default) or `flex` containers. The flexbox layout makes the buttons as wide as the text for each class and automatically wraps row-wise. The grid layout has a custom number of columns, which can be adjusted using the *Number of*

**Figure 5** **Three common types of sound event detection and classification.** (A) The most common setup where an entire audio clip is labelled either using binary (*e.g.*, bird/not bird) or multi-class classification. It does not give any explicit temporal resolution; (B) displays sound event detection (SED). Labels typically have a fixed width (time duration), but the resulting automatic detection models classify successive time windows (small fractions of the file duration) for each class, with consecutive windows belonging to the same class being joined together to form a larger sound event. In the above example, the blackbird classification is longer than the other two detections. This illustration does not include any frequency resolution (the annotations span the entire *y*-axis) but a similar setup would have a small number of frequency bins where a species is expected to occupy a given range; (C) is object detection setup using bounding boxes. This clearly shows highest resolution detail among the annotation examples shown, with the possibility of overlapping sound events in both time and frequency domains. In this toy example, the two wren annotations have differing frequency ranges which gives richer frequency information. Adapted from *Stowell (2022)*.

*Columns* parameter in **Other Settings**. The grid layout is neater while the flexbox layout is more compact. The differences can be seen in Fig. 6.

The different groups of classes are colour coordinated to distinguish the main classes, miscellaneous categories and manually added classes. Core classes in the selected site species list have green borders, while classes manually added to the list are in blue. Miscellaneous sounds such as human, insect or weather noise are outlined in orange. If a label is present in the file that is not in the collective class list, its bounding box is coloured grey in the plot.

When a new class is added to the list, the list is searched and if it is not already present, the class is appended. If the class is found, an error message is thrown saying the class is already present in the list.

The British Trust for Ornithology has a list of two-letter species codes for over 250 bird species. The app has the ability for users to view classes in the displayed list with their corresponding code, if they have a matching species in the BTO list. `bto_codes.csv` acts as a lookup table, which was sourced from www.bto.org/sites/default/files/u16/downloads/forms_instructions/bto_bird_species_codes.pdf. It has two columns, **bto_code** and **species_name**: the first column has the two letter code associated with the species name in the second column. An example of the conversion is shown on Fig. 6.

## Other workflow features

At the top left of the screen, there is a hidden/collapsed sidebar that contains multiple adjustable drop-down menus of settings. These include **Configuration:** general setup inputs for the annotation project, such as the directory containing the audio files, uploading more files to the user's folder, uploading annotations from previous projects and the column of `species_list.csv` with the relevant classes for labelling, which is expanded upon in Extension to other audio labelling projects. **Sound Settings:** adjustable inputs for processing incoming audio files, including dB gain (or amplification factor) and the length of audio

**Figure 6 Class list layouts.** Shows the possible class list layouts for a small example list of classes, as well as the miscellaneous categories and one manually added class. (A) The default layout of the classes in a grid, with the number of columns decreased to three for display purposes. (B) is the same list in flexbox layout. Note the reduced amount of empty space from shorter class names. (C) shows the same flexbox layout with class names changed to their corresponding BTO codes, where applicable. The space is compacted further which becomes more apparent with longer class lists.

clips to display in the spectrogram (the default is 15 s). **Spectrogram Settings:** adjust the visual aspects of the spectrogram plot, *e.g.*, the colour palette, plot height and whether to include time and frequency guidelines for boxes drawn. **FFT Settings:** parameters for the Short-Time Fourier Transform (STFT) calculation via the spectro function. **Other Settings:** miscellaneous widgets for adjusting the class list, label summary table and label edit table.

Metadata is a helpful addition to annotation workflows, providing context for users who may not have been involved in the project design or data collection phases. The app attempts to link audio files to a predefined list of recorders that were deployed in order to give information to end-users on aspects of the study site such as habitat type, location and distance to coastline, then displays them in a collapsible panel to be consulted as desired. For example, the location of the recordings may give important context for detecting species, in particular for migratory birds. Addressing the information needs of labellers by providing all available metadata for context can improve the accuracy of the classifications (*Mortimer & Greene, 2017*).

With a key objective of the app being labeller efficiency, using keyboard shortcuts instead of repetitive mouse drags and clicks enables increased precision and productivity. This was not included in the main procedure but is a useful addition to efficient labeller workflow. Available hotkeys include saving and deleting labels, navigating to the previous or next file, pausing and playing the current audio file and adding or deleting classes in the class list.

The label edit table is an interactive revision tool that describes all labels for the current file. This can be enabled in the Other Settings tab in the sidebar, where it then appears at the bottom of the app. It will only display when the current file has at least one label. Some of the fields from saved annotations can be edited, such as the time/frequency limits of the bounding boxes, class identified and label confidence.

The summary table is another revision tool providing an abridged description of all files in the workflow folder. It can also be enabled through the Other Settings tab. The user can filter these files by name, number of labels present per file and even split the label counts

per file by the classes present. Users can quickly navigate between a large number of files to find other examples of previously labelled species.

When the labelling project is complete and the user wishes to access their labels (especially on a server), they can be exported using the Download button. This is located in the **Label buttons** section towards the bottom of the main body of the app. These labels are exported to the Downloads folder of the user's local machine. This download feature works both on a server or local deployment.

## CASE STUDY

All examples and results in this paper come from data recorded for the Nature+Energy project (*MaREI, the SFI Research Centre for Energy, Climate and Marine, 2022*), part of which involves developing an acoustic monitoring system for on-shore wind farm sites in Ireland. Over 1,700 h of audio was collected between April and July 2022. Wildlife Acoustics Song Meter Mini Bat (https://www.wildlifeacoustics.com/products/song-meter-mini-bat) units were mounted at approximately 1.5 metres above the ground, placed between a wind turbine and some linear feature such as hedgerows or gaps in tree cover. For the first stage of the project, we selected five recording sites (one recorder per location) representing the variety of habitat types present.

Nature+Energy is a collaboration between academic researchers and industry partners that aims to measure and enhance biodiversity at onshore wind farms throughout Ireland. The project focuses on exemplar wind farm sites, located in typical habitats where wind farms are situated in Ireland.

Wind energy is undergoing substantial growth in Ireland, providing over 30 percent of its energy needs in 2020 and expected to reach 80 percent by 2030 (*Department of the Environment, Climate and Communications; Department of the Taoiseach, 2019*). With the increase in wind turbines comes the possibility of biodiversity degradation, with bird and bat populations being particularly vulnerable to collision mortality (*Richardson et al., 2021*; *Choi, Wittig & Kluever, 2020*). The need for monitoring systems is crucial to inform habitat management planning, potentially reducing habitat decline, fatalities and issues specific to each site. Through the Nature + Energy project, a number of acoustic sensors were set up as a non-invasive monitoring method at select study sites across the country. A description of each recorder, and the sites where each was deployed, is given in Table 2.

Passive acoustic monitoring provides rich insight into species abundance, and temporal and spatial granularity (*Warren et al., 2021*). Due to the nature of where the recorders are located, anthropogenic noise from the wind turbines themselves has an adverse effect on data quality, especially for labelling species with calls occupying lower frequencies. There is a need for frequency granularity to conceal other species or noise sources occurring simultaneously. Metadata, as mentioned in Other workflow features, is a helpful addition to give labellers insight on aspects of the study site such as habitat type, time of day/year, and location.

**Table 2  Wind farm study site info.**

| Site | Recorder | Location | Land use type | Area (Ha) | No. Turbines | Hours recorded Acoustic | Ultrasonic |
|---|---|---|---|---|---|---|---|
| Carnsore Point | RECORDER | Wexford | Agricultural (coastal, pasture) | 80 | 14 | 547 | 19.6 |
| Cloosh Valley | CLOOSHVALLEY | Galway | Commercial forestry on peat substrate | 1378 | 36 | 438 | 11.9 |
| Rahora | RAHORA | Kilkenny | Agricultural (inland, arable) | 24 | 5 | 170 | 14.2 |
| Richfield | RICHFIELDM1 | Wexford | Agricultural (coastal, arable) | 112 | 18 | 850 | 52.4 |
| Teevurcher | TEEVURCHER | Meath | Agricultural (inland, pasture) | 60 | 5 | 156 | 1.7 |

## Project annotation procedure

A version of the app was deployed to an Rstudio Connect server (*RStudio Team, 2020*) integrating user login via Auth0 (https://auth0.com/). Users (bird experts) were assigned a sample of audio files from multiple wind farm sites that were believed to contain sounds of interest. A summary of the actions involved in labelling a selected segment of audio with a bounding box (Steps ③ to ⑥ in Detailed labeller instructions) is outlined in Fig. 7 below.

## Results of labelling

A small sample of the results obtained from labelling are shown in Fig. 8. These examples illustrate the diversity of bird vocalisations found on wind farms, even within the same species. The structure, duration and frequency ranges of bird vocalisations are shown to vary, the latter of which could not be extracted via one dimensional labels. Noise from the wind turbines on the study sites is especially noticeable in the examples on the left.

A sample of the count data for non noise-related classes labelled are shown in Table 3. There are several zeroes in the table, indicating the diversity of the selected study sites, which cover different geographic regions and habitat types. Further details on the bird species identified by bird experts, using NEAL, are during the case study are included in Table 4 in the bird conservation status in the appendix. It displays bird conservation data and information on their distribution across Ireland, and was obtained from *Gilbert, Stanbury & Lewis (2021)*.

Some common species such as the Blackbird, Meadow Pipit and Robin are present across almost all sites, while the Yellowhammer is only present at one site due to it being a rare species that feeds on cereal crops. The annotations that were discerned to be birds but could not specifically be identified (representing about 8% of non-noise related labels collected, not included in the table) can be returned to the same labeller for deeper analysis or given to another labeller for a second opinion. Both could make use of the output of a trained species recognition machine learning model to inform the decision.

Counts of annotations from the first set of data gathered from deployed recorders are shown in Fig. 9. It displays the activity periods of selected species from the recordings. The recorders were deployed from April-July, which in Ireland is a period when day length is at its greatest; on June 21st, the summer solstice and longest day of the year, Ireland receives approximately 17 h of daylight. This figure provides an example of the type of data that can
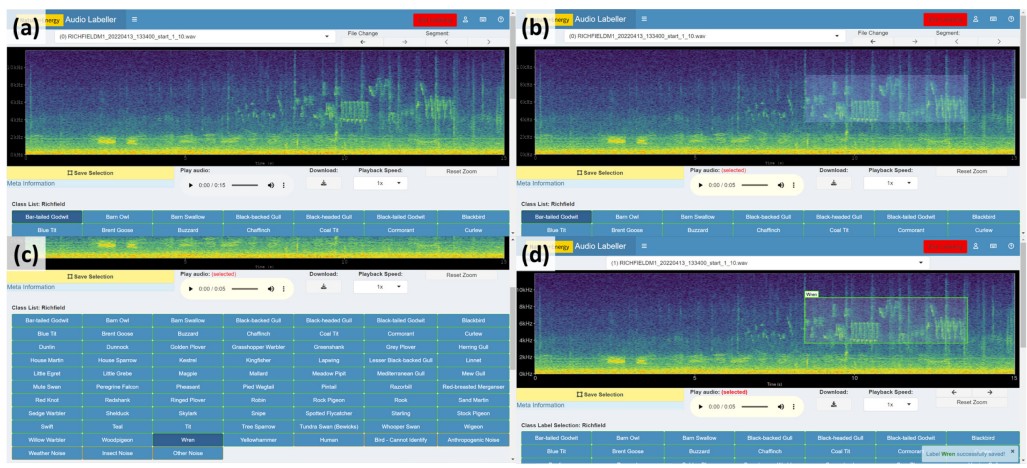

**Figure 7** **Spectrogram labelling example.** (A) View spectrogram and play audio. (B) Drag a box around the audio of interest. (C) Select identified class. (D) Click Save Selection.

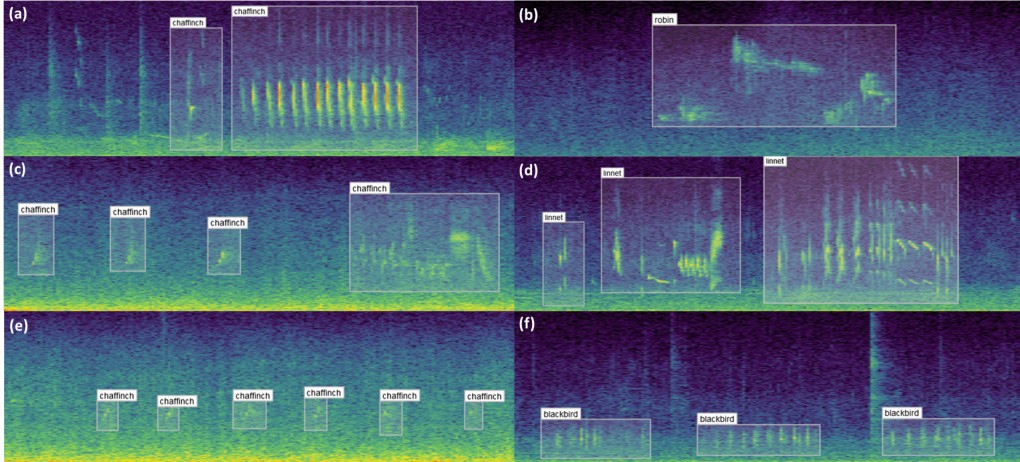

**Figure 8** **Example bounding box annotations.** Example annotations of sample audio collected from wind farm sites across Ireland. (A), (C) and (E) in the first column show different vocalisations of the chaffinch. (A) can be readily identified both visually and through listening to the audio, whereas (C) and (E) are increasingly difficult to distinguish from noise. Adjusting the contrast or applying noise reduction in the spectrogram settings may help with identifying examples similar to these. The structure of the vocalisations in (A) and (C) are quite dissimilar despite coming from the same species, which can be optionally captured using the call_type input. (B) shows a distinct robin vocalisation which was easily identified given little background noise. (D) includes three separate vocalisations of a linnet, since the sounds are separated by a natural pause. (F) contains three vocalisations of a blackbird. This sits partially in the range of wind turbine noise, and would be difficult to identify if the noise was more prominent or the bird was located further from the recorder.

be extracted from acoustic recordings, and it could potentially be used to aid management decisions, especially for particularly vulnerable species such as raptors. Some species such as the Yellowhammer tend to be identified throughout the day, whereas others such as the

**Table 3  Example raw species counts from labelled data.** The table details the most and least common species identified by NEAL. The top species were those with a total count of more than 10, while less frequent species are shown below. Some bird species, such as the Robin (*Erithacus rubecula*), regularly exploit a wide variety of habitats and food resources, and thus are common and widely distributed. Others however, have more specific habitat requirements. Oystercatchers (*Haematopus ostralegus*), for example, are limited to coastal habitats where they feed on large invertebrates such as mussels. NEAL identified several birds of conservation concern in Ireland (Status **Red**), including the Yellowhammer, Curlew, and Meadow Pipit.

| Class | Carnsore point | Cloosh valley | Rahora | Richfield | Teevurcher | Conservation status |
|---|---|---|---|---|---|---|
| Blackbird | 37 | 0 | 107 | 73 | 96 | Green |
| Chaffinch | 0 | 141 | 0 | 32 | 0 | Green |
| Dunnock | 26 | 1 | 0 | 12 | 0 | Green |
| Goldfinch | 12 | 0 | 0 | 0 | 0 | Green |
| Hooded Crow | 0 | 0 | 13 | 1 | 16 | Green |
| House Sparrow | 0 | 0 | 20 | 0 | 0 | Amber |
| Linnet | 49 | 0 | 15 | 42 | 2 | Amber |
| Meadow Pipit | 17 | 1 | 33 | 15 | 51 | Red |
| Pied Wagtail | 7 | 0 | 1 | 4 | 2 | Green |
| Robin | 0 | 62 | 4 | 49 | 7 | Green |
| Rook | 0 | 0 | 10 | 83 | 0 | Green |
| Sedge Warbler | 11 | 0 | 0 | 1 | 0 | Green |
| Starling | 0 | 0 | 37 | 0 | 0 | Amber |
| Stonechat | 30 | 0 | 0 | 0 | 0 | Green |
| Wren | 26 | 0 | 12 | 143 | 0 | Green |
| Yellowhammer | 0 | 0 | 60 | 0 | 0 | Red |
| Blue Tit | 0 | 0 | 0 | 7 | 0 | Green |
| Buzzard | 0 | 0 | 2 | 0 | 1 | Green |
| Coal Tit | 0 | 6 | 0 | 1 | 0 | Green |
| Curlew | 1 | 0 | 0 | 0 | 0 | Red |
| Goldcrest | 0 | 3 | 0 | 0 | 0 | Green |
| Grasshopper Warbler | 3 | 0 | 0 | 0 | 0 | Amber |
| Great Tit | 0 | 0 | 0 | 5 | 0 | Green |
| Jackdaw | 0 | 0 | 0 | 0 | 1 | Green |
| Magpie | 0 | 0 | 1 | 0 | 0 | Green |
| Mallard | 3 | 0 | 0 | 1 | 0 | Green |
| Oystercatcher | 1 | 0 | 0 | 0 | 0 | Amber |
| Pheasant | 0 | 0 | 1 | 0 | 0 | Green |
| Skylark | 1 | 0 | 0 | 0 | 0 | Amber |
| Song Thrush | 3 | 0 | 0 | 0 | 0 | Green |
| Swallow | 1 | 0 | 0 | 0 | 9 | Amber |
| Teal | 0 | 1 | 0 | 0 | 0 | Amber |
| Woodpigeon | 0 | 0 | 4 | 1 | 0 | Green |

Robin and Chaffinch occur in shorter bursts of activity interspersed with lulls. If multiple recorders were placed on the sites, it could also inform how bird species are using the site throughout the day—for example, to determine whether they are just passing through the study area or are foraging multiple locations throughout the day.

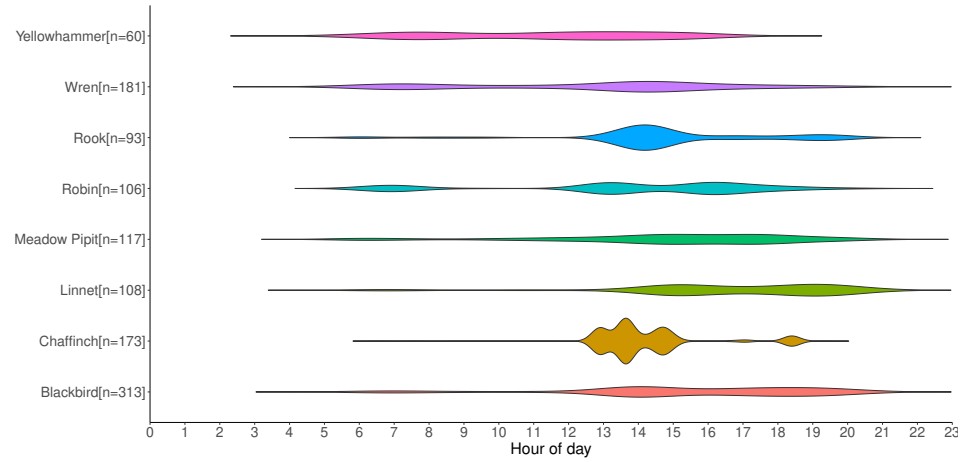

**Figure 9** Distribution of selected labelled species by timestamp in recording.

## EXTENSION TO OTHER AUDIO LABELLING PROJECTS

Our app is built so that it can be configured for other audio labelling tasks beyond the bird species classification demonstrated in our case study. Here we provide some guidance on setting up the input files if they are being expanded to similar areas or changed to cover other domains.

To proceed, a new user merely needs to download (or clone) the GitHub repository. Upon running, the app automatically will create a new user if one is not found in the www directory. The user can then add audio files into the XXX directory. The expected audio filename format is `RECORDERNAME_YYYYMMDD_HHMMSS.wav`. If the files are split up into smaller segments, the additional convention `_start_MM_SS` should be appended before the filename extension to indicate the number of minutes/seconds into the recording that they start. This allows the date and time to be parsed from the above filename format and shown in the metadata panel.

Users can copy annotations (or upload if using on a server) from previous labelling projects using the upload button in the Configuration tab in the sidebar panel. These labels could be from manual work or generated by an ML model or some other automatic method. The CSV label data should at least contain the following:

- **date_time:** Date and time label was made in ISO 8601 format (`yyyy-mm-ddTHH:MM:SS+HHMM`)
- **file_name:** Name of the corresponding audio file in the Data Folder
- **start_time:** Start time (in seconds) of the bounding box
- **end_time:** End time (in seconds) of the bounding box
- **start_freq:** Start frequency (in kHz) of the bounding box
- **end_freq:** End frequency (in kHz) of the bounding box
- **class_label:** Class of sound (*e.g.*, bird species) assigned to this bounding box

Labels will be stored in `labels/labels_<username>.csv` if such a folder by this name exists in the above directory. Otherwise the labels are saved in `labels/labels_tmp.csv`. The information contained in each label includes: the time the annotation was made; for which audio file; the start and end time/frequency bounds of each bounding box; the class name; label confidence; labeller and optional call type and additional notes.

The app uses multiple read-only input data tables to accompany the folder of audio files. These include the recorder (study site) location data, species lists and BTO codes. If any of these are edited or replaced for a new annotation project the files should be located in the parent folder of the GitHub repository, and saved in `.csv` format using `UTF-8-BOM` encoding. This ensures that column names and text strings are read correctly.

NEAL allows for multiple species lists to be included, all of which are stored in `species_list.csv`. The file has one column for each site being studied, with the first entry of each column being the site name. Columns can have different numbers of rows (species), *i.e.*, all those one expects to find at the given site. The species names can be stored in the column in any order as they are sorted alphabetically at run time. Columns can be appended to it by uploading via a widget in the Configuration tab. Each of the columns in the uploaded data not already present in the species data are appended.

The `location_list.csv` file has a list of recorders that were deployed along with accompanying metadata about the sites studied. If the current audio file in the above format matches a recorder in the list, the following columns of the file are passed to the metadata tab:

- **recorder_name:** prefix for audio file names recorded by this device
- **lat:** latitude of the recorder for the study period, in decimal degrees
- **long:** longitude of the recorder for the study period, in decimal degrees
- **location_name:** name of the study site
- **location_county:** county of study site
- **habitat_type:** primary habitat type of the study site
- **dist_to_coastline:** approximate distance to the nearest coastline in kilometres

If any of this information is unavailable, the column name in the metadata panel will still appear but the body of text will be blank. Extra columns can be added to the file, where they will be printed verbatim to the metadata panel.

## CONCLUSION

NEAL is an open-source application for visually examining and annotating audio data. The tool was designed with the primary goal of improving labeller efficiency and consistency. Vocalisations are tagged with comprehensive annotations providing time and frequency detail, accurate to several decimal places. Call type and an open text field for notes attempt to capture multiple modes of information, with the possibility of performing multiple labelling tasks simultaneously. A labeller providing classifications of different bird species, as well as the call type and identifying sources of noise in each sound clip, has generated three potentially separate sets of labels without expending more effort than generating only one. These can each serve as targets for a machine learning algorithm to predict.

The Shiny package for R allows for an interactive front-end, while its reactive ability reduces unnecessary computational expense. Its no-code interface allows domain experts to interact with the data without any knowledge of R programming. While the app was designed mainly for classifying bird audio, it can be expanded to projects with data focusing on bats, frogs, small mammals and insects—all are popular in bioacoustics research.

There is room for extending the functionality of the app. Expanded contextual information such as weather data, proximity to Special Areas of Conservation and habitat type could prove particularly useful. New interactive tables and visualisations (even the static summary visualisations presented in this paper) could be used to investigate outliers and labeller inconsistency. The app being open-source means others can contribute or request features, driving innovation for future releases. R is one of the most popular programming languages in bioacoustics and new features to the app can be easily added on through the Shiny app's modular design.

While we did spend a significant amount of time during the development of the app to reduce the rendering time for the spectrogram plots, we could not find a feasible way to do get to the speed of the compared-with apps in R which would also produce high resolution interactive plots. Future versions of the app hope to find methods to speed up computation to more acceptable levels. Since the app uses `ggplot`, which we have found to be slow in rendering a large number of tiles for the spectrograms, NEAL's performance should improve in line with its development. If the app continues to lag behind the compared-with software, some `C++` plugins may be able to help with render speeds.

# APPENDIX A. DETAILED LABELLER INSTRUCTIONS

A more detailed version of the general workflow defined above in General labelling workflowcarried out by labellers is below. Items without numbers are optional steps.

① Go to the deployed app at the given URL. Login via the Auth0 page.

② Click the user icon in the top right, then start labelling. The first audio file and corresponding spectrogram should load

- Nagivate to the next file with little or no annotations. This can be inspected with the dropdown menu, displaying the number of annotations in brackets.

- Tune parameters in the sidebar to the desired configuration. In particular, the spectrogram parameters such as colour palette and contrast may be adjusted until the user is comfortable with the visual distinction of the sounds present.

- If the user can already visually detect sound events (*e.g.*, bird vocalisations or common forms of noise), these can be annotated as described in steps ③-⑥.

③ Play the audio until the user comes across a sound of interest. Once identified, place the mouse at the top left corner of the vocalisation you identified and drag to the bottom right of the object, drawing a moderately tight box around it.

④ The audio player now updates to have a **filtered audio file**, reconstructed using only the times and frequencies within the box drawn. This should assist the user in identifying the sound by removing much of the noise present from other frequencies, and cropping

the audio to the small clip of interest. To return to the raw audio file, click anywhere on the plot.

⑤ Once they have identified sound, if it is in the class list. If it is present, click the class button; otherwise add it using the text box below and add it to the list.

- If any additional information can be extracted from the audio, such as the call type of the species or miscellaneous notes, or the labeller's confidence of a particular annotation is less than certain, they can be included using the text fields on the bottom right of the app's main page.

⑥ When the desired class is selected from the class list, redraw a tight box around the vocalisation and click **Save selection** directly below the plot. This will save the annotation and draw a bounding box with the chosen label onto the plot.

- If the user wishes to change any of the bounding boxes, they can be deleted using the **delete** button in the **label buttons** section and and redrawn using the above steps. Alternatively they can be edited using the label edit table which can be enabled in the **Other Settings** tab. This is described more in the Extension to other audio labelling projects section.

⑦ **Continue annotating** the file by repeating steps ③ to ⑥ until no more unlabelled sounds of interest remain.

⑧ **Proceed to the next file** using the file navigation menu or adjacent navigation buttons and go back to step ③.

⑨ Once the user finishes their session, click the user icon again, followed by **End Labelling** to finish.

## APPENDIX B. BIRD CONSERVATION STATUS

## ACKNOWLEDGEMENTS

We thank our labellers Harry Hussey, David Kelly, Seán Ronayne and Mark Shorten, who annotated the majority of the audio files, as well as providing beta-testing of the app in its early stages. We also thank the on-site personnel and surveyors for carrying out recorder maintenance, *i.e.*, data and battery transfers, as the wind farms are widely spread across Ireland. Finally, we would like to acknowledge the significant work done by Dr. Aoibheann Gaughran on the Nature + Energy project, and for her encouragement and comments on the Shiny app from its inception. This paper is dedicated to her memory.

**Table 4 Bird species identified with conservation status and distribution across Ireland.**

| Common name | Scientific name | Bird family | Conservation status | Ireland distribution |
|---|---|---|---|---|
| Blackbird | Turdus merula | Thrushes | Green | Widespread and common resident |
| Blue Tit | Cyanistes caeruleus | Tits | Green | Widespread and common resident |
| Buzzard | Buteo buteo | Raptors | Green | Widespread and common resident |
| Chaffinch | Fringilla coelebs | Finches | Green | Widespread and common resident |
| Coal Tit | Periparus ater | Tits | Green | Widespread and common resident |
| Curlew | Numenius arquata | Waders | Red | Declining resident population & winter visitor to wetlands |
| Dunnock | Prunella modularis | Dunnocks | Green | Widespread and common resident |
| Goldcrest | Regulus regulus | Kinglets | Green | Common resident (coniferous forests) |
| Goldfinch | Carduelis carduelis | Finches | Green | Widespread and common resident |
| Grasshopper Warbler | Locustella naevia | Warblers | Amber | Widespread summer visitor |
| Great Tit | Parus major | Tits | Green | Widespread and common resident |
| Hooded Crow | Corvus cornix | Crows | Green | Widespread and common resident |
| House Sparrow | Passer domesticus | Sparrows | Amber | Widespread and common resident |
| Jackdaw | Corvus monedula | Crows | Green | Widespread and common resident |
| Linnet | Carduelis cannabina | Finches | Amber | Widespread and common resident |
| Magpie | Pica pica | Crows | Green | Widespread and common resident |
| Mallard | Anas platyrhynchos | Ducks | Green | Common resident (wetlands) |
| Meadow Pipit | Anthus pratensis | Pipits | Red | Common resident (rough pastures and uplands) |
| Oystercatcher | Haematopus ostralegus | Waders | Amber | Resident & winter visitor (coast and inland lakes) |
| Pheasant | Phasianus colchicus | Game Birds | Green | Introduced- Widespread and common resident |
| Pied Wagtail | Motacilla alba yarrellii | Wagtails | Green | Widespread and common resident |
| Robin | Erithacus rubecula | Chats | Green | Widespread and common resident |
| Rook | Corvus frugilegus | Crows | Green | Widespread (absent from expansive uplands) |
| Sedge Warbler | Acrocephalus schoenobaenus | Warblers | Green | Widespread and common summer visitor |
| Skylark | Alauda arvensis | Skylarks | Amber | Common resident (uplands and areas of farmland) |
| Song Thrush | Turdus philomelos | Thrushes | Green | Widespread and common resident |
| Starling | Sturnus vulgaris | Starling | Amber | Widespread and common resident |
| Stonechat | Saxicola rubicola | Chats | Green | Widespread resident (scrubland, mainly near the coast) |
| Swallow | Hirundo rustica | Swallows & Martins | Amber | Common summer visitor |
| Teal | Anas crecca | Ducks | Amber | Small numbers throughout Ireland |
| Woodpigion | Columba palumbus | Pigeons & Doves | Green | Widespread and common resident |
| Wren | Troglodytes troglodytes | Wrens | Green | Widespread and common resident |
| Yellowhammer | Emberiza citrinella | Buntings | Red | Declining resident mainly in the east and south. Strongly tied to cereal cultivation. |

### Funding

This publication was produced by the Nature+Energy Project, funded by Science Foundation Ireland (12/RC/2302_P2) and MaREI, the SFI Research Centre for Energy, Climate and Marine Research and Innovation, with additional funding from Microsoft and the SFI CONNECT Centre for Future Networks and Communications (13/RC/2077_P2). In addition, Andrew Parnell's work was supported by: a Science Foundation Ireland Career Development Award (17/CDA/4695); SFI Centre for Research Training in Foundations of Data Science 18/CRT/6049, and SFI Research Centre awards I-Form 16/RC/3872 and Insight 12/RC/2289_P2. The funders had no role in study design, data collection and analysis, decision to publish, or preparation of the manuscript.

### Grant Disclosures

The following grant information was disclosed by the authors:
Science Foundation Ireland: 12/RC/2302_P2.
MaREI, the SFI Research Centre for Energy, Climate and Marine Research and Innovation.
Microsoft.
The SFI CONNECT Centre for Future Networks and Communications: 13/RC/2077_P2.
A Science Foundation Ireland Career Development Award: 17/CDA/4695.
SFI Centre for Research Training in Foundations of Data Science: 18/CRT/6049.
SFI Research Centre awards I-Form: 16/RC/3872, Insight 12/RC/2289_P2.

### Competing Interests

The authors declare that there are no competing interests.

### Author Contributions

- Anthony Gibbons conceived and designed the experiments, performed the experiments, analyzed the data, prepared figures and/or tables, authored or reviewed drafts of the article, and approved the final draft.
- Ian Donohue conceived and designed the experiments, performed the experiments, analyzed the data, prepared figures and/or tables, authored or reviewed drafts of the article, and approved the final draft.
- Courtney Gorman analyzed the data, authored or reviewed drafts of the article, and approved the final draft.
- Emma King analyzed the data, authored or reviewed drafts of the article, and approved the final draft.
- Andrew Parnell conceived and designed the experiments, performed the experiments, analyzed the data, prepared figures and/or tables, authored or reviewed drafts of the article, and approved the final draft.

### Data Deposition

The NEAL (Nature+Energy Audio Labeller) is an open-source interactive audio data annotation tool and available on GitHub: https://github.com/gibbona1/neal.

The audio data and labels available at https://github.com/gibbona1/neal_data.

## Supplemental Information

Supplemental information for this article can be found online at http://dx.doi.org/10.7717/peerj.15913#supplemental-information.

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
