# Peer review of "NEAL: an open-source tool for audio annotation"

_PeerJ, doi:10.7717/peerj.15913_

## Round 0.1 · original submission · Minor Revisions

Both reviewers appreciated your work,and they make constructive comments that should improve the final version. This is nice contribution to the software section of the journal.

Reviewer 1 ·

Basic reporting

Good English, good references.

Some problems with the lay-out:
(1) the first author's name appears twice in the title;
(2) missing opening parentheses in all references in the text, e.g. "ecological monitoring Ross and Allen (2014)" instead of e.g. "ecological monitoring (Ross and Allen 2014)"; this occurs with every reference throughout the text;
(3) perhaps related to the previous: p.3 "R packages, such as Attali (2021); Bailey (2022);..." Those are authors' names, not names of packages.

Experimental design

I think that description of software falls within the scope of this journal ("BioInformatics Software tools").

It is mainly explained clearly, although only at the very end ("upload") I realized that the standard workflow involves uploading data to a server, so it seems to be a web app, although it is claimed that the app can work locally. This could be made clearer.

p.2 "We present the NEAL app" sounds strange, after already having introduced the exact same thing as "the Shiny app". Probably a copy-paste thingy.

To replicate, I would like to know what "istft" is doing. Presumably this is some sort of vocoder; is the Griffin-Lim algorithm used, or something with better quality? Perhaps you are keeping the complex-valued spectral slices?

Date-time format is ambiguous w.r.t. time zone and daylight saving time. Such details are relevant in bioacoustics. Why not ISO 8601?

Validity of the findings

The authors make a big deal out of the refreshing-strategies of the app. But other state-of-the-art software is *very* fast at rendering full-screen spectrograms (much faster than running the spectral analysis), so the authors might consider computing the visual spectrogram pixel by pixel instead of as 720,000 tiles.

Additional comments

Not so clear: "the phase of the sounds outside the selection are collapsed to zero". Can you elaborate why this is so?

Cite this review as

·

Basic reporting

The authors have developed an open-source application named NEAL to aid in the efficient and consistent examination and annotation of audio data. The tool enables detailed annotations of vocalisations in terms of time and frequency, vocalisation type, and an open text field for notes, thereby capturing multifaceted information. This makes the application rich in terms of its utility for simultaneous multiple labelling tasks, and each of these labels could serve as a target for machine learning algorithms. The tool leverages the Shiny package for R, providing an interactive front-end with reactive functionality, without requiring knowledge of R programming from the users. Despite its primary design for bird audio, it can be extended to other bioacoustics taxonomies. The authors also envision potential extensions of the app to include more contextual data and novel visualisations, and its open-source nature enables other researchers to contribute to its development. I am very happy to read this paper and authors have done a great job at describing the features and usability of the application. Here are some minor comments I would like to bring to the author's notice:

1. While the authors accurately point out that high-quality annotations produced by experts are essential for training supervised learning algorithms, it is important to clarify that not all machine learning algorithms fall under the supervised learning category. Indeed, there are self-supervised and unsupervised learning algorithms as well. Therefore, a more accurate phrasing could be, "These models, when operating as supervised learning algorithms, require training on high-quality annotations produced by experts." This maintains the authors' original intent while accommodating the broader scope of machine learning methodologies.

2. On line 328 and again in the caption of the Table 3, the authors employ the phrase, "species identified by NEAL," to actually refer to annotations made using NEAL. 'Species recognition' has a very specific meaning in audio-ML and should not be conflated with in this manner.

3. The authors have made elaborate comparisons between NEAL and several other labelling software, but one notable omission is 'Sonic Visualiser,' a longstanding tool widely employed for similar tasks. I am wondering if there is a specific reason why 'Sonic Visualiser' wasn't included in the authors' comparative analysis.

Experimental design

no comment

Validity of the findings

no comment

---

## Round 0.2 · accepted · Accept

Thanks for providing detailed answers to all reviewers' comments and revising the manuscript accordingly. This is an app that I hope will be used by e.g. the increasing number of ecologists using acoustic methods. The manuscript is now ready for publication.

---

## Author Rebuttal · Round 0.2

Dear Professor Yoccoz,

We would like to thank you for reviewing our paper titled *NEAL: An open-source tool for audio annotation* for publication in the in the software section of PeerJ.

We would especially like to thank the reviewers for their helpful comments and suggestions on the paper. We have addressed them in detail below. In particular we have corrected the citation style, added another application to the comparison table, improved clarity on the inner workings of the audio processing in the app, reworded some sentences to be more explicit in their explanations, and included more details on the limitations of the app.

We hope that you now find the manuscript suitable for publication in PeerJ.

Yours sincerely,

Anthony Gibbons

On Behalf of all authors.

# Reviewer 1

## Basic reporting

> (1) the first author's name appears twice in the title

We have corrected this typo, it came from accidentally uncommenting a line in the LaTeX document prior to submitting.

> (2) missing opening parentheses in all references in the text, e.g. "ecological monitoring Ross and Allen (2014)" instead of e.g. "ecological monitoring (Ross and Allen 2014)"; this occurs with every reference throughout the text

Many thanks for spotting this. We have changed all relevant occurrences of `\cite{}` to `\citep{}`

> (3) perhaps related to the previous: p.3 "R packages, such as Attali (2021); Bailey (2022);..." Those are authors' names, not names of packages.

We have cleared up the language here. The new citation commands hopefully make this clearer than it was before.

## Experimental design

> It is mainly explained clearly, although only at the very end ("upload") I realized that the standard workflow involves uploading data to a server, so it seems to be a web app, although it is claimed that the app can work locally. This could be made clearer.

We agree that it is worth clarifying to avoid confusion here. NEAL is primarily intended to be run locally in R, but can also be deployed to a server (we have clarified this in line 67 of the paper). Towards the end of the paper where we mention "uploading", we emphasise that users can upload if running on a server (as a web-app), or just copy files to the given location if using locally. Navigating to different directories works the same locally and online.

> p.2 "We present the NEAL app" sounds strange, after already having introduced the exact same thing as "the Shiny app". Probably a copy-paste thingy.

We have changed the wording here to emphasise NEAL is a Shiny App, and thank the reviewer for pointing out this inconsistency.

> To replicate, I would like to know what "istft" is doing. Presumably this is some sort of vocoder; is the Griffin-Lim algorithm used, or something with better quality? Perhaps you are keeping the complex-valued spectral slices?

We have included a sentence to state that ISTFT makes use of the complex values of the spectrogram to reconstruct filtered audio. We have also included an extra reference so that the reader can obtain more details should they wish.

> Date-time format is ambiguous w.r.t. time zone and daylight saving time. Such details are relevant in bioacoustics. Why not ISO 8601?

We thank the reviewer for spotting this. The datetime to which you are referring, i.e. the time each label is created, now uses ISO 8601 format.

## Validity of the findings

> The authors make a big deal out of the refreshing-strategies of the app. But other state-of-the-art software is *very* fast at rendering full-screen spectrograms (much faster than running the spectral analysis), so the authors might consider computing the visual spectrogram pixel by pixel instead of as 720,000 tiles.

This is another good point on the speed of loading spectrograms in the app. In order to produce images to load pixel by pixel, the app would first have to create the spectrogram image, which would take the same amount of time as rendering it to the screen currently. It is also important that the plot be tied to the spectrogram data so the bounds of the selected area can be found (both for filtering audio and drawing bounding boxes). Using ggplot in R makes this connection automatic, but we have found this library to be slow. We have addressed this issue in a new paragraph in the conclusion as possible future work in later releases of NEAL.

## Additional comments

> Not so clear: "the phase of the sounds outside the selection are collapsed to zero". Can you elaborate why this is so?

We apologies for not making this clearer. We had meant "the magnitude of the spectrogram values outside the selection are collapsed to zero", i.e. the complex values in the spectrogram which are used to reconstruct the audio with only the selected area of interest being audible. We have changed *phase* to *magnitude* on line 209.

# Reviewer 2 (Burooj Ghani)

## Basic reporting

> 1. While the authors accurately point out that high-quality annotations produced by experts are essential for training supervised learning algorithms, it is important to clarify that not all machine learning algorithms fall under the supervised learning category. Indeed, there are self-supervised and unsupervised learning algorithms as well. Therefore, a more accurate phrasing could be, "These models, when operating as supervised learning algorithms, require training on high-quality annotations produced by experts." This maintains the authors' original intent while accommodating the broader scope of machine learning methodologies.

This is certainly a useful suggestion and we thank the reviewer for the contribution. We have changed the paragraph as suggested and believe it improves the clarity of that section.

> 2. On line 328 and again in the caption of the Table 3, the authors employ the phrase, "species identified by NEAL," to actually refer to annotations made using NEAL. 'Species recognition' has a very specific meaning in audio-ML and should not be conflated with in this manner.

Many thanks for this comment. We have changed the text to emphasise that these are species identified by bird experts using NEAL, not the outputs of a species recognition model.

> 3. The authors have made elaborate comparisons between NEAL and several other labelling software, but one notable omission is 'Sonic Visualiser,' a longstanding tool widely employed for similar tasks. I am wondering if there is a specific reason why 'Sonic Visualiser' wasn't included in the authors' comparative analysis.

We had come across this software but initially did not include it as it seemed quite similar to Audacity. After this comment however, we have now included it

in the table alongside the other software for comparison. We thank the reviewer for spotting this omission.